# Testing the Conjecture That Quantum Processes Create Conscious Experience

**DOI:** 10.3390/e26060460

**Published:** 2024-05-28

**Authors:** Hartmut Neven, Adam Zalcman, Peter Read, Kenneth S. Kosik, Tjitse van der Molen, Dirk Bouwmeester, Eve Bodnia, Luca Turin, Christof Koch

**Affiliations:** 1Google Quantum AI, Los Angeles, CA 90291, USA; viathor@google.com; 2Read Family Foundation, Penn HP10 8LL, UK; 3Neuroscience Research Institute, Department of Molecular, Cellular and Developmental Biology, UC Santa Barbara, Santa Barbara, CA 93106, USA; kosik@lifesci.ucsb.edu (K.S.K.); tjitse@ucsb.edu (T.v.d.M.); 4Department of Physics, UC Santa Barbara, Santa Barbara, CA 93106, USA; bouwmeester@ucsb.edu (D.B.); ebodnia@ucsb.edu (E.B.); 5Huygens-Kamerlingh Onnes Laboratory, Leiden University, 2311 EZ Leiden, The Netherlands; 6Faculty of Medicine and Health Sciences|Biomedical Research, University of Buckingham, Buckingham MK18 1EG, UK; luca.turin@buckingham.ac.uk; 7Allen Institute, Seattle, WA 98109, USA; christofk@alleninstitute.org

**Keywords:** physical substrate of consciousness, quantum biology, brain–computer interface, brain organoids, anesthesia, xenon

## Abstract

The question of what generates conscious experience has mesmerized thinkers since the dawn of humanity, yet its origins remain a mystery. The topic of consciousness has gained traction in recent years, thanks to the development of large language models that now arguably pass the Turing test, an operational test for intelligence. However, intelligence and consciousness are not related in obvious ways, as anyone who suffers from a bad toothache can attest—pain generates intense feelings and absorbs all our conscious awareness, yet nothing particularly intelligent is going on. In the hard sciences, this topic is frequently met with skepticism because, to date, no protocol to measure the content or intensity of conscious experiences in an observer-independent manner has been agreed upon. Here, we present a novel proposal: *Conscious experience arises whenever a quantum mechanical superposition forms.* Our proposal has several implications: First, it suggests that the structure of the superposition determines the qualia of the experience. Second, quantum entanglement naturally solves the binding problem, ensuring the unity of phenomenal experience. Finally, a moment of agency may coincide with the formation of a superposition state. We outline a research program to experimentally test our conjecture via a sequence of quantum biology experiments. Applying these ideas opens up the possibility of expanding human conscious experience through brain–quantum computer interfaces.

## 1. A Conjecture Inspired by Roger Penrose

In 1989, in his seminal book “The Emperor’s New Mind”, Roger Penrose made an intriguing proposal [1]. He suggested that quantum processes are essential in forming the physical substrate of consciousness. This idea is attractive because the equations of quantum mechanics tell us that at any moment in time, an object, myself or the world at large, exists in a superposition of many configurations. Yet, in any given moment, we only experience one. To illustrate this, imagine a researcher who steps up to one of the quantum computers in Google’s Quantum AI lab to observe a quantum bit prepared in a superposition of two states |0〉 and |1〉. If the researcher sees the qubit in state |0〉, then the Schrödinger equation, which governs the time evolution of quantum systems, tells us that there is another version of the researcher that sees the qubit in state |1〉. This feature of quantum physics—that parallel worlds are spawning all the time—has perplexed physicists ever since quantum physics was invented. Hugh Everett, in his 1957 doctoral thesis, was the first to notice that the equations of quantum mechanics, if taken literally, describe a multiverse composed of co-existing parallel worlds [2]. In quantum physics, one refers to the world of our everyday experience as ‘classical’. If we accept a model of reality in which we are living in a multiverse, then an attractive explanation of consciousness becomes available: *Consciousness is how we experience the emergence of one unique classical reality from the many the multiverse is composed of.*

Penrose’s proposal was very definite. He suggested that a conscious moment occurs whenever gravity induces a quantum mechanical superposition to collapse, a process he named ‘objective reduction’. However, here, we argue that his original proposal is in need of refinement.

First, we want to remove the reference to gravity. Whether gravitational influences cause a collapse is testable, but so far, it has not been observed in a lab, and initial experiments looking for such an effect have come up short [3]. We prefer to stay with textbook quantum mechanics, which teaches that an ‘effective collapse’ occurs whenever a quantum system interacts with an external system with a large number of degrees of freedom, as, for instance, happens during a measurement of a quantum system by a macroscopic system, such as an instrument or a person. This interaction causes ‘environment-induced decoherence’, i.e., the quantum system of interest becomes entangled with the degrees of freedom of the external system, which, from the vantage point of the quantum system of interest, looks like an effective collapse of the superposition. This explains why we can predict classical probabilities for measurement outcomes in a specific basis, set by the coupling to the external system (the environment). Quantum superposition for the entire system (the system of interest plus the environment) still persists. The notion of collapse merely indicates that while the system of interest may still exist in a superposition of classical states, the superposition can no longer be observed via interference. In other words, the system of interest evolved from a pure quantum state into a mixture of classical states that coexist without knowing of each other’s existence.

Second, *we propose that a conscious moment occurs whenever a superposition forms, not when it collapses.* This modification is necessary to avoid the possibility of faster-than-light communication that Penrose’s original proposal entails when applied to multiple entangled qubits, as explained below. It also sidesteps the conceptual problem of having to define when exactly a measurement occurs, as all interactions are being treated alike. So our proposal is firmly rooted in Everett’s ‘many worlds’ formulation of quantum mechanics. Figure 1 depicts an example quantum circuit in which gate and measurement operations act on three qubits, each of which is initialized in the |0〉 state. A superposition is formed in the first qubit, then it spreads across all qubits via entangling operations and finally ‘effectively’ collapses, or, following Penrose, truly collapses by some real physical process, when one of the qubits is measured in the |0〉, |1〉 basis.

In Roger Penrose’s original proposal, a conscious moment occurs whenever a superposition collapses. However, if, as in this example, the superposition involves more than one qubit, then the question arises as to how the experience is distributed across the qubits when a first qubit is measured. There are two possible answers: (i) Only the qubit that is measured first, and causes the superposition to collapse, has an experience. (ii) All three qubits experience something. Answer (i) runs into difficulties because in a relativistic setting, the order in which the three qubits are measured depends on the reference frame. This leads to the unsatisfying conclusion that whichever qubit has a conscious experience also depends on the reference frame. (One could evade this difficulty by conjecturing that whenever a qubit is measured, a conscious moment arises, regardless of its quantum state. However, since this conjecture holds for classical states without superpositions as well, it amounts to saying that a conscious moment occurs whenever one particle bumps into another.) Answer (ii) is problematic as well because if all three qubits experience a conscious moment, then this could be used to construct a communication channel, for instance, by implementing a Morse code, which would enable the transmission of information faster than the speed of light. Recall that in standard quantum mechanics, it is well established that entanglement cannot be employed to achieve faster-than-light communication. It is the assumption that a collapse leads to a conscious experience across the spatially separated qubits that leads to the violation.

To avoid this violation, we posit that a moment of consciousness arises not when a superposition collapses, but when it forms. The structure of the evolving superposition and the paths the experiencing quantum system follows within it determine the qualities of the conscious experiences, “what it is like to be” that system in that state [4,5]. Our key tenet is that a system only ever experiences classical, *definite* states. These states form a basis of the Hilbert space the quantum system lives in. If a classical state evolves into a quantum superposition through a unitary transformation, as prescribed by the Schrödinger equation, then the experiencing system will evolve along multiple paths simultaneously. Each path, connecting basis states, is known as a ‘Feynman path’ and results in a distinct sequence of conscious moments. Hence, we posit that a quantum system may be composed of many experiencing minds, albeit often very simple ones [6,7]. As the system evolves from a basis state it just experienced, the population (probability mass), initially concentrated on this start state, will spread to other basis states within a light cone given by the prevailing interactions. The more the population decreases on the start state, the more likely a transition becomes. Eventually, depending on which Feynman path the state is moving along, the system will experience the transition to a different basis state. To illustrate how in our proposal the formation of a superposition creates a moment of consciousness, let us consider several examples (see Table 1).

Since quantum superposition is a basis dependent notion, we need to address the question of how a preferred basis is selected. Here, we can follow proposals made by David Deutsch or by Wojciech Zurek, who suggested that the interaction between subsystems selects a preferred basis, referred to alternatively as the ‘interpretation basis’ by Deutsch [8] or the ‘pointer state basis’ by Zurek [9]. See also Wallace for a monograph on this topic [10]. It may also be possible to apply a principle from Integrated Information Theory and choose the basis that maximizes the integrated information the system experiences [11,12,13].

In our proposal, entanglement solves the binding problem, that is, the unity of a phenomenal experience [14,15], in a natural manner and without any additional assumptions. The number of degrees of freedom involved in the entanglement bound the amount and content of consciousness [16]. To quantify the content of an experience in a principled manner, the Φ-measures proposed by a quantum mechanical instantiation of integrated information theory seem advantageous, such as Zanardi et al. in 2018 [11,12] or Albantakis et al. in 2023 [13]. The experimental tests described in the next section are capable of probing the various hypotheses inspired by Penrose’s ideas, including his original one.

## 2. A Sequence of Quantum Biology Experiments

A key challenge when attempting to construct a scientific theory of consciousness is that conscious experience is not a traditional experimental observable with an associated objective, that is, subject-independent measurement protocol. Indeed, as famously argued by Descartes, my experience is the only thing I can be absolutely certain of [17]. Whether anything else is conscious, such as you, has to be inferred as the most likely explanation of all available information. It follows that I cannot conclusively demonstrate that anyone except myself is conscious. That is, epistemological solipsism, the position that I can only be certain of my own consciousness, is logically consistent [18]. Still, inferring consciousness in others is common and implicit—as in asking people what they see or how they feel. In neurologically impaired patients, such as in those with disorders of consciousness, speaking is replaced by command following (e.g., move your eyes). Without such communication, detecting consciousness becomes challenging [19].

To measure the position or velocity of an object like an airplane, any engineer knows what to do. But when tasked with answering to what degree the autopilot of the airplane, a large language model, such as Google’s Gemini, or a brain organoid, is conscious, we do not agree on how to go about measuring this. Without either a well-accepted general theory of consciousness or an experimental test, any such claims remain vacuous. As some of us have argued in a recent article [20], consciousness does become an observable in the limit case that the observing system and the observed systems are exact copies of one another. But to the degree the system we are observing is dissimilar to ourselves, this protocol breaks down, hence the unresolved debate to what degree another person, a fish, a tree, a rock or a computer executing an LLM may be conscious. Here, we describe another approach that allows us to provide experimental evidence that a theory of consciousness is accurate. It relies on expanding one’s own consciousness; hence, we will refer to this method as the *expansion protocol*. It sidesteps the challenge that has plagued consciousness research. In particular, it would allow us to test whether the creation of a superposition generates a conscious experience.

### 2.1. Expanding the Mind Using Quantum Processors

In an experimentum crucis, one would establish a physical link between a human brain and a quantum computer that would enable coherent interactions and mediate entanglement. If our conjecture is accurate, this should enable richer conscious experiences of the combined system, requiring more descriptive bits than the experiences the human reports without the link. Moreover, such a setup may enable dialing in specific qualia of an experience, as explained in the next paragraph. Figure 2 depicts coupling a system in a quantum mechanical superposition state residing in a subject’s brain $|\psi_{Me}\rangle$ with a set of qubits in superposition inside a quantum computer $|\psi_{Chip}\rangle$. Before the systems are coupled, their respective states exist in separate state spaces, known as Hilbert spaces, of dimension *N* and *M*, respectively. After they are made to interact, the wave function describing the combined system $|\psi_{Cyborg}\rangle$ resides in an N×M-dimensional Hilbert space. We conjecture that a superposition forming in this higher dimensional state space would be experienced by the subject as a richer experience as compared to a superposition state forming in the lower *N*-dimensional Hilbert space describing the isolated brain of the subject. A tantalizing conjecture is that this sort of expanded consciousness occurs during psychedelic, mystical, near-death and other types of extraordinary experiences [21] not by coupling to an external system, but by increasing the number of entangled qubits participating in superspositions forming in the brain.

Likewise, if we were to stay with Roger Penrose’s original proposal, it stands to reason that the collapse of a quantum mechanical superposition into a classical state requiring more bits to describe creates a richer experience than one collapsing into a classical state characterized by fewer bits.

While such experiments are possible in principle, they are technologically very challenging and may require invasive technologies. Today’s technologies to establish brain– computer interfaces are not designed to support a coherent coupling (one that allows the spreading of entanglement) between structures in a brain and a quantum computer. Any such link would have to occur without perturbing the human brain sitting inside its protective skull and membranes.

But future quantum sensing technologies, such as nitrogen vacancy probes combined with optogenetic methods, may eventually allow for such protocols. A myriad of details would need to be engineered, i.e., which cell structures in which nerve cells in which parts of the brain are most suitable, and to which degrees of freedom in nervous tissue should one couple, and how. In short, we need to state what constitutes the neurobiological qubits. While various proposals have been made, such as nuclear spins (Fisher [22]), collective modes in microtubules (Penrose and Hameroff [23]) or aromatic rings (Hameroff [24]), we refrain from committing at this point and prefer to approach answering this question experimentally.

First, we suggest conducting a preparatory experiment. It consists of mediating entanglement between two room temperature qubits via a biological substrate. It is based on a theoretically well-founded approach to elucidate quantum mechanical properties of unknown systems. It considers three systems, two of which are well characterized and can be thought of as qubits or ensembles of qubits, referred to as $Q_{1}$ and $Q_{2}$. A system of interest, in our case a biological substrate B, is introduced. We then couple $Q_{1}$ coherently to B, and likewise, $Q_{2}$ is coherently coupled to B, taking care not to establish a direct link between $Q_{1}$ and $Q_{2}$. If we then find that it is possible to mediate entanglement between $Q_{1}$ and $Q_{2}$ via the system B, then we can conclude that B requires a quantum mechanical description [25,26]. This paradigm is rather flexible, as different coupling schemes and different substrates B can be investigated. For instance, B could be a microtubule, a receptor protein such as rhodopsin, a single nerve cell or even a brain organoid. This protocol is illustrated in Figure 3.

### 2.2. Turning the Mind off Using Isotopes of Xenon

In order to decide which biological structures are the most promising candidates to select as system B and how to implement the coherent coupling, we want to start with an even simpler experiment. In fact, this experiment, to some extent, has already been performed. In 2018, Li and co-workers exposed four groups of mice to four different isotopes of xenon that differ in the number of neutrons in their nucleus [27]. Xenon is an inert noble gas with anesthetic properties [28]. Indeed, xenon (with an atomic number of 54), has many properties that make it, in some sense, an ideal inhalation agent—it is odorless, nontoxic, nonexplosive, environmentally friendly and, due to its chemical stability, does not transform in the body [28]. Intriguingly, the work of Li et al. on the anesthetic effects of xenon isotopes shows that the potency of the two isotopes with half-integer nuclear spin (Xe129 and Xe131, spin of 1/2 and 3/2, respectively) is about 30% less than the potency of the two isotopes with zero nuclear spin (Xe132 and Xe134). This difference, if confirmed, cannot be explained by differences in the outer electron shells (there are none) and is unlikely to be caused by differences in atomic mass (<1% difference between Xe131 and Xe132). If true, the results suggest that some of the effects of xenon on consciousness may be mediated by nuclear spins, quantum systems amenable to superposition. It is suggestive that the half-integer spin isotopes were the ones with reduced anesthetic potency. The non-zero spin may act as a qubit contributing to the formation of larger superpositions, which, according to our conjecture, would be correlated with a richer conscious experience counteracting the anesthetic effect.

Li et al. studied the ED50 of loss of the righting reflex of 80 mice treated with the different xenon isotopes (20 mice per isotope), which provides little information on the mode of action of xenon. Given that two out of three experiments in the biomedical sciences are not reproducible [29], it is important to replicate important experiments, in particular if they challenge widely held assumptions. Thus, we plan to repeat this experiment, but this time, we want to ‘anesthetize’ brain organoids, focusing on achieving results with high statistical confidence. Brain organoids, carefully instrumented to allow for high bandwidth measurements via high density electric or optical arrays, like those in the Kosik and Bouwmeester lab at UC Santa Barbara, have emerged as a potent platform to investigate living nervous tissue that share characteristics with human brain tissue (see Figure 4) [30].

As a complementary study to establish the veracity of behaviorally relevant xenon isotope effects, we also plan to administer the xenon isotopes to an invertebrate, the fruit fly, Drosophila Melanogaster, with about 100,000 neurons. Xenon does not fully anesthetize fruit flies at ambient pressure. We plan to measure the pressure at which its isotopes cause immobility. This will allow us to complement the organoid study with behavioral data with good statistics, as we can apply the anesthetic gasses to a sufficiently large number of flies to obtain statistically highly reliable results. The experiment can be conducted within an electronic spin spectrometer, and spin resonance will be employed to monitor spin signals. This part of the study will be conducted in the lab of Luca Turin at the University of Buckingham [31].

If we can replicate the findings by Li et al. in these two, quite different, model nervous systems, we are in a position to discover the underlying cellular processes. The electromagnetic field emanating from a nuclear spin is rather weak compared to the electromagnetic field generated by an electron. In brain tissue, there are plenty of overlapping electromagnetic fields generated by moving charges, electron, nuclear or orbital spins, or by external sources, such as the earth magnetic field or cell phone towers. We already have an indication from the Meyer–Overton law that anesthetics act in lipid, hydrophobic pockets [32]. (The Meyer–Overton rule states that the potency of an anesthetic can be predicted by its solubility in olive oil [33].) So a rather specific cellular environment is required for the weak isotopic spin effects to manifest. For instance, it has been proposed that this involves a radical pair mechanism, a quantum biological effect that is suspected to underlie bird magnetoreception [34] or lithium’s effects on hyperactivity [35,36,37]. In conclusion, the necessary level of specificity may provide a clue as to which process relevant to sustaining the physical correlate of consciousness is interrupted during anesthesia, and in turn, this may suggest how to couple a qubit to implement the expansion protocol.

## 3. Dialing in Qualia with a Brain-Quantum Computer Interface

Our conjecture, if correct, suggests a novel approach to understand the relationship between states of matter and experience. If we would be successful in implementing the expansion protocol, we could more precisely investigate these correlations. Imagine asking a user with a brain linked to a quantum processor about their feelings and subsequently measuring the qubits’ states. Then, one would look for correlations between the valence of the reported emotions and the bit string measured. This could reveal the qualia associated with different states of matter. Consequently, we may gain the ability to tailor experiences with unprecedented detail and precision.

Before achieving brain–quantum computer coupling, let us consider a simpler question: can volunteers given sub-anesthetic doses distinguish different xenon isotopes, say, by their differential psychedelic properties? While the answer remains unknown [38], there is evidence of isotope effects influencing human sensing and cognition. For instance, humans can differentiate the smell of odorants with hydrogen atoms replaced by deuterium [39], and they can taste heavy water, with hydrogen atoms replaced by deuterium, as sweeter than regular water [40]. Lithium 6 and lithium 7 exhibit different effects on mood [41]. Phosphorous nuclear spins have been proposed by Matthew Fisher as one of the most viable candidates for qubits in the brain that may confer quantum algorithmic advantages [22,42]. However, in none of these proposals have differential mass effects been ruled out. One reason xenon is attractive is that it is heavier than the isotopes used in the experiments cited, making it less likely that mass effects can account for the observed differences among isotopes. These observations suggest the potential significance of spin-dependent mechanisms in biology, opening up a new field we may call *biological spintronics*. In the coming years, we expect advancements in attaching spin markers to specific molecules within cellular tissues, granting us unprecedented control over biological processes at the molecular level in real-time [43].

In summary, we are proposing a fundamental research program to uncover whether quantum effects are underlying the physical substrate of consciousness. Central to this endeavor is the establishment of coherent coupling between quantum degrees of freedom in brain tissue and a quantum processor. Utilizing modern quantum biological methods, we aim to achieve this coupling in a non-invasive manner (i.e., without surgical intervention). If this program were to be successful, then it would allow for building technical aides that could expand human conscious experience in space, time and complexity (see Figure 5).

## 4. Formation of Quantum Superposition May Facilitate Agency

The formation of a superposition state may not only create a conscious experience, but may go hand-in-hand with a moment of agency. Maybe an organism can exercise some degree of choice in the libertarian sense over which classical configuration it is going to experience next. Whether Nature takes advantage of this possibility, we cannot yet say. What we can say for certain is that an external observer, in general, will not be able to predict the behavior of a living organism, not even probabilistically. If the number of quantum bits describing the evolution of a quantum system approaches about 100 (which is less than needed to accurately describe the electronic structure of a protein or the chromodynamics inside a proton), then, in general, it will be the case that the observer cannot even give probabilities anymore with which certain outcomes will occur. This is because computing the probabilities of the $2^{100}$ outcomes exceeds the capabilities of any computing machine that can be constructed [20]. This situation is referred to as *Knightian uncertainty* [44].

But why should we suspect that behind this veil of unpredictability moments of agency are hiding? Our argument rests on the following observation: *behaviors conducive to our well being, i.e., conducive to maintaining homeostasis, tend to be correlated with feelings of pleasure, while actions that threaten homeostasis feel unpleasant*. We call this the ‘homeostatic correlation’. If humans were adequately described as deterministic automata, then this homeostatic correlation is difficult to explain. If behaviors were predetermined, then evolution would not select for the homeostatic correlation, as it would not matter how an organism feels since its behaviors are determined anyway. Giving the automaton access to a source of randomness to select behaviors does not change the fundamental argument since, in this case, it does not matter either how the organism feels. It is instructive to consider two behaviors, both essential to the survival of a species, having sex and giving birth. The former is typically associated with pleasure, the latter with pain. Why is this so? We would argue that the former involves choice while the latter does not, at least during prehistoric times, when most of human evolution took place. So it seems to us that Nature uses feelings as a lure or a deterrent, but this only works if an organism can act to attain pleasant and avoid unpleasant states. A possible way to explain the correlation between pleasant sensations and behaviors conducive to an organism’s well-being is to postulate the existence of agency [12,45]. If an organism possesses the agency to freely choose a state, then presumably, it would choose emotionally rewarding over unrewarding ones. Evidence for this could be found by using the expansion protocol. We could measure $|\psi_{Cyborg}\rangle$ and ask the user about the valence of their emotions. If we observe a higher occurrence of pleasant than unpleasant states, this may indicate that the user has the agency to influence the outcome. Quantum operations, such as creating or collapsing superposition states, may give a system (experiencing a single classical reality at a given time) the freedom to exert agency and to express a preference. In this view, selecting a single classical configuration from the multitudes contained in a superposition implements conscious experience (of that classical configuration) and gives the system agency to select it.

In conclusion, we argue that the operations available to a quantum processor may be necessary to implement sentience and agency. Vice versa, today’s AI systems running on semiconducting electronics are confined by the laws of classical information theory. Their computations can be abstracted by the operations of a probabilistic Turing machine. If the above arguments are correct, it follows that these operations are insufficient to implement consciousness and agency. Stated more pointedly, *Turing machines have become intelligent but may never become conscious*. For the latter, a quantum Turing machine is required.

## Figures and Tables

**Figure 1 entropy-26-00460-f001:**
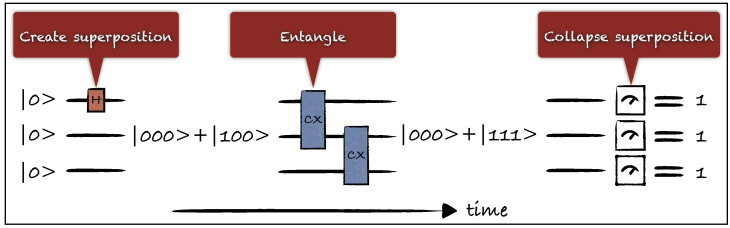
A generic quantum circuit consisting of gate and measurement operations acting on three qubits. All qubits are initialized in the |0〉 state. A Hadamard gate creates a superposition on the first qubit. Controlled X gates then entangle the first qubit with both the second and the third qubit, generating the superposition |000〉+|111〉. At the end of the circuit, each qubit is measured, effectively collapsing the superposition into |111〉. With which operation shall we identify the physical correlate of consciousness? We suggest a conscious moment occurs when a superposition forms.

**Figure 2 entropy-26-00460-f002:**
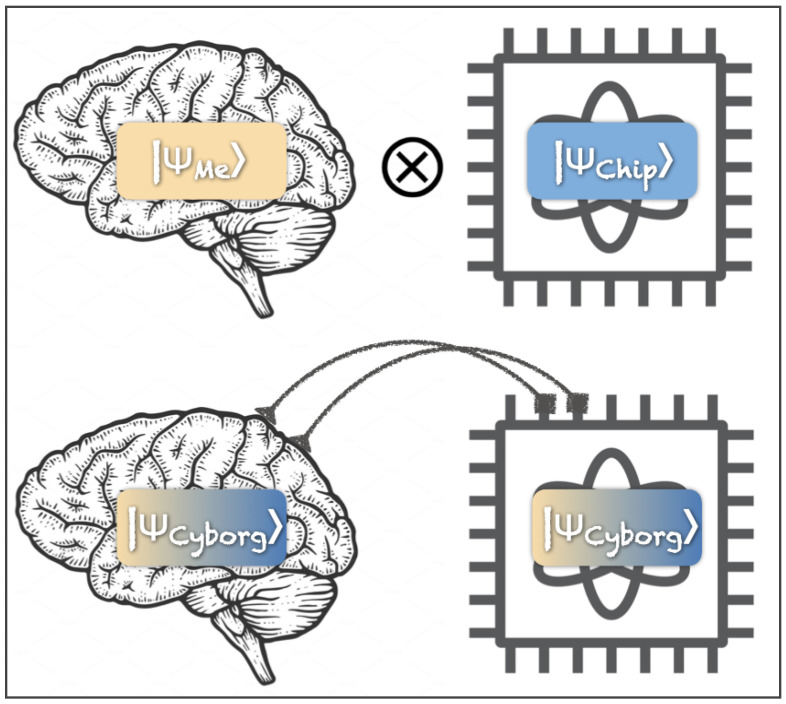
Conceptual depiction of a coherent coupling between a human brain and a quantum processor. If our conjecture is accurate, then we should be able to generate richer experiences by creating the superposition $|\psi_{Cyborg}\rangle$, which describes the combined system and which lives in a larger state space (Hilbert space) as compared to $|\psi_{Me}\rangle$ and $|\psi_{Chip}\rangle$, which describes the separate systems.

**Figure 3 entropy-26-00460-f003:**
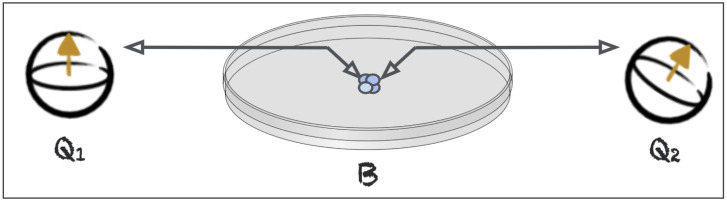
Two qubits, $Q_{1}$ and $Q_{2}$, linked coherently (as indicated by the horizontal arrows) via a biological substrate B. This setup allows us to learn whether B can act as a quantum channel and mediate entanglement between $Q_{1}$ and $Q_{2}$.

**Figure 4 entropy-26-00460-f004:**
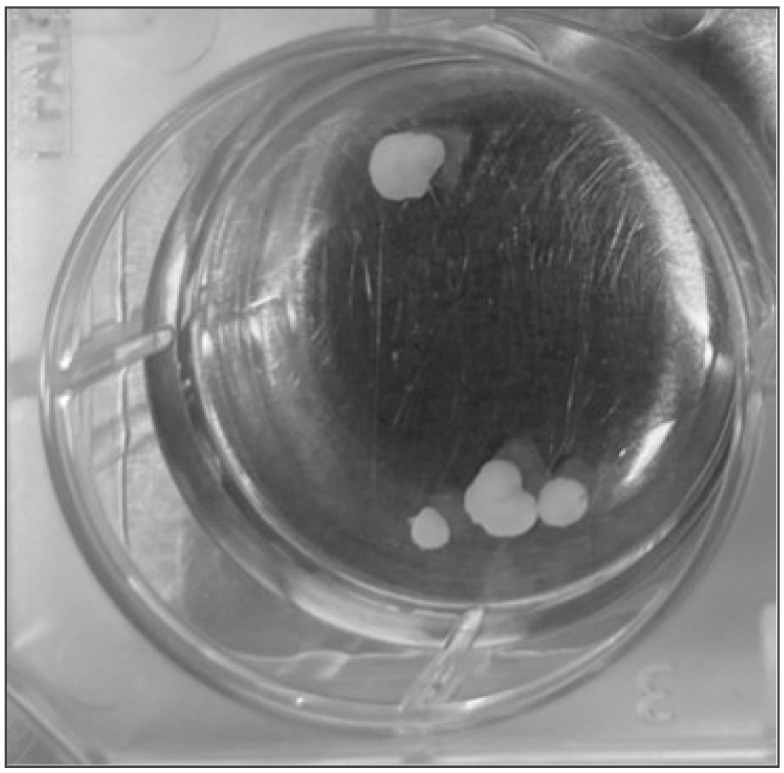
Cerebral organoids derived from human-induced pluripotent stem cells can model human brain development in anatomical organization, cellular composition and physiological signaling. Brain organoids, here about 2–3 mm in diameter, containing about 100,000 neurons, exhibit structured, non-random and fast electrical activity (including action potentials). As such, the organoid electrical activity represents a generic human brain. Photo courtesy of Kenneth Kosik.

**Figure 5 entropy-26-00460-f005:**
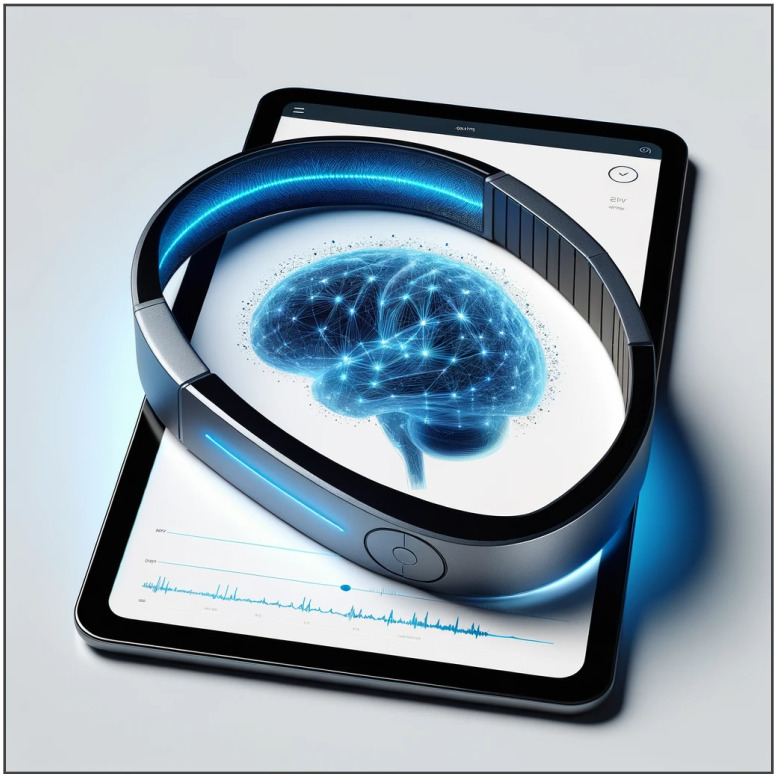
Coherently linking a brain to an external quantum processor, ideally by employing non-invasive techniques, would allow for building technical aides that could expand human conscious experience in space, time and complexity. Rendering by DALL·E 2.

**Table 1 entropy-26-00460-t001:** We assert that at any given moment, a system can only experience definite classical states. Accordingly, when a superposition forms, the system consciously experiences only its constituent basis states. To illustrate this, we enumerate conscious experiences that arise when the classical |000〉 state undergoes quantum mechanical transitions to other states. Here, the assumption is that the computational basis, consisting of tensor products of |0〉s and |1〉s, single qubit states, is the preferred basis.

State Transition	Associated Experience
|000〉→|000〉	No experience
|000〉→|111〉	No experience
|000〉→α|100〉+β|010〉+γ|001〉	Experiences either 100 or 010 or 001. The modulus square of the probability amplitude determines the probability that the corresponding configuration is experienced.
|000〉→|0〉(α|10〉+β|01〉)	Experiences either 10 or 01 on the second and third qubit.
|000〉→(α|0〉+β|1〉)(γ|10〉+δ|01〉)	Two separate, non-integrated experiences. The first is either 0 or 1 and the second is 10 or 01.
|000〉→(α|0〉+β|1〉)(γ|0〉+δ|1〉)(ϵ|0〉+ζ|1〉)	Three separate, non-integrated experiences. Each consists of experiencing either 0 or 1.

## Data Availability

No new data were created or analyzed in this study. Data sharing is not applicable to this article.

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
