# Peer review of "Testing the Conjecture That Quantum Processes Create Conscious Experience"

_entropy, 2024, doi:10.3390/e26060460_

Round 1

Reviewer 1 Report

Comments and Suggestions for Authors

Neven et al. here hypothesize that conscious experience arises whenever a quantum mechanical superposition forms. The idea appears to exhibit some similarity to Penrose's conceptualization of consciousness but is distinct insofar as it posits that conscious moments occur whenever a superposition forms, not when it collapses (or undergoes orchestrated objective reduction as in Penrose's proposal). The authors argue that this modification is necessary to avoid the possibility of faster-than-light communication ostensibly enabled by entanglement. However, this concern appears ill-founded, as it is now well established that the existence of entanglement, even if abstaining from the local realism that many desire, does not violate special relativity through enabling superluminal communication. This is so, as the quantum mechanical measurements are inherently probabilistic, which in combination with the no-cloning theorem and no-communication theorem precludes the faster-than-light transmission of information through entanglement that the authors allude to and use as motivation for their hypothesis. What they appear to miss is that the correlation as encoded in entangled state is not an interaction and thus cannot be used to transfer information unless combined with a classical channel. It appears to me that this invalidates the authors' argument in favor of associating consciousness with the formation of superpositions. I also note that the corresponding section in the manuscript is entirely unreferenced.

The authors suggest that the experiencing system only ever experiences classical states. They suggest that it does so by choosing a classical path to follow the evolution/generation of superpositions, which results in conscious moments. It is left open how this coupling could be achieved. The authors exclude wavefunction collapses (if accepting the Copenhagen interpretation) and, I assume, decoherence (as the Everettian equivalent; its disturbingly unclear which notion the authors prefer, as the introduction alludes to many worlds while the rest repeatedly refers to wavefunction collapse), based on the above discussion. Thus, it appears to me that through their postulate the authors grant the non-local wave function itself a real character attribute. This and equating consciousness with the formation of coherences appears questionable, as the mere formation is preceding interactions with the system and the concept is basis-dependent, whereby the "preferred" basis might not be established, or multiple preferred bases (coherent and incoherent ones) might be equally relevant if several mutually exclusive "following paths" existed. For example, consider a double-slit experiment as a form of proxy of a consciousness-generating neuronal quantum process. Referring the two outcomes, particle passed through slit 1 and 2, as |1> and |2>, respectively, the particle will be in state 1/sqrt(2)(|1> + |2>) prior to passing the slits, thereby, according to the authors potentially contributing to a conscious experience. However, how could the state prior to its collapse/interaction/etc. feedback to the experiencing system and do so in a way that even permits agency through mere existence? It seems to me that the authors proposition "conscious experience arises whenever a quantum mechanical superposition forms" violates causality. What if another processes utilizes a "rotated" analyzer, thereby measuring alternative superpositions of states, thus proposing an alternative decomposition of the initial state, e.g. in the form of |1> = 1/2(|1> + |2>) + 1/2(|1> - |2>) = 1/sqrt(2)(|a> + |b>)? Dose this state, coherent in |a>,|b> but incoherent in |1>, contribute to a conscious experience, or not? With these fundamental questions unexplained, or at least incomprehensible to me, I do not see how the proposition can be considered publication ready. Please expand on the fundamental idea beyond keywords.

The authors suggest a range of quantum biological experiments, which sound interesting, not necessarily to support their hypothesis, but on the simpler level of exploring if quantum entanglement plays a role in consciousness. The idea appears to be to expand the mind through coupling to a quantum computer and turning the mind off using general anesthesia. While the idea is nice, the practical realizability appears to be entirely left out. Still, the suggestion is formulated as a grant proposal (which, so I assume, was the original purpose of this manuscript), as if this could be attempted tomorrow. However, details or concrete ideas are glaringly absent.  In any case, it appears that the coupling to a quantum devise can only be successful if the neuronal qubits can couple to the device. This however requires a focus on a specific realization of qubits, which the authors leave open. Nuclear spins are mentioned, but then electron spin resonance is suggested to be employed and radical pairs are mentioned, thereby putting focus on electron spins. The manuscript would profit greatly if the authors explored possible implementations of their idea in terms of the various suggestions in the literature (or alternatives). It is also unclear how Xe would suppress the generation of superpositions, but at the same time enrich the conscious experience by contributing additional qubits if the nuclear spin is non-zero.

The authors assume that a quantum system comprising about 100 qubits will preclude the prediction of behavior. While their argument is true in a simplistic way, such large systems will increasingly lose their quantum traits and appear classical to the observer, permitting a treatment in terms of classical statistical physics. In fact, even the much smaller biological qubits that have been suggested so far lose coherence/entanglement on timescales that are fast relative to human cognition/perception and do so the fast the larger they are. This suggests that truly we are at most able to consider independent local systems of a few degrees of freedom. Assuming that these systems signal "0" or "1" in some form with equal probability, almost all outcomes will have a fraction 0.5 of qubits being “1”, permitting a classical model.  

Author Response

We want to thank the reviewer for taking the time to study our paper and for their thoughtful comments. Addressing those comments will certainly improve our article.

Neven et al. here hypothesize that conscious experience arises whenever a quantum mechanical superposition forms. The idea appears to exhibit some similarity to Penrose's conceptualization of consciousness but is distinct insofar as it posits that conscious moments occur whenever a superposition forms, not when it collapses (or undergoes orchestrated objective reduction as in Penrose's proposal). The authors argue that this modification is necessary to avoid the possibility of faster-than-light communication ostensibly enabled by entanglement. However, this concern appears ill-founded, as it is now well established that the existence of entanglement, even if abstaining from the local realism that many desire, does not violate special relativity through enabling superluminal communication. This is so, as the quantum mechanical measurements are inherently probabilistic, which in combination with the no-cloning theorem and no-communication theorem precludes the faster-than-light transmission of information through entanglement that the authors allude to and use as motivation for their hypothesis. What they appear to miss is that the correlation as encoded in entangled state is not an interaction and thus cannot be used to transfer information unless combined with a classical channel. It appears to me that this invalidates the authors' argument in favor of associating consciousness with the formation of superpositions. I also note that the corresponding section in the manuscript is entirely unreferenced.

We understand that in textbook quantum mechanics entanglement cannot be employed to enable superluminal communication. (Several of us frequently teach this foundational fact to students and other audiences :)) However, Penrose’s proposal goes beyond textbook quantum and by doing so he re-opens this issue.

Let us get back to our example in Figure 1. We prepare a three qubit Greenberger Horne Zeilinger state and then consider measurements, possibly on all three  qubits. Therefore the state collapses (already by the first of the measurements) . To satisfy Penrose and for argument’s sake we can even admit that the collapse was gravity induced because the measurement device might have displaced sufficient mass to trigger the Diosi criterion. So according to Penrose a moment of consciousness occurred. Now we can ask the question: Which of the three qubits experienced a conscious moment? There are two possible answers: 1) Only the qubit that was measured first has an experience. 2) All three qubits experience something (even if only one qubit was measured). 

Answer 1) runs into difficulties, because in a relativistic setting the order in which the three qubits are measured depends on the reference frame. This leads to the unsatisfying conclusion that which qubit has a conscious experience also depends on the reference frame. Answer 2) is problematic because if all three qubits experience a “bing” moment, as Stuart Hameroff would call it, then this could be used to construct a communication channel, say by implementing a Morse code, which would enable faster than light communication.

To clarify this further we added text in the section starting at line 70.

The authors suggest that the experiencing system only ever experiences classical states. They suggest that it does so by choosing a classical path to follow the evolution/generation of superpositions, which results in conscious moments. It is left open how this coupling could be achieved. The authors exclude wavefunction collapses (if accepting the Copenhagen interpretation) and, I assume, decoherence (as the Everettian equivalent; its disturbingly unclear which notion the authors prefer, as the introduction alludes to many worlds while the rest repeatedly refers to wavefunction collapse), based on the above discussion. 

We are firmly placing our conjecture within the Everett many-worlds account of quantum mechanics and the manuscript says so explicitly. Whenever we use the term collapse we mean “apparent collapse” or “effective collapse” involving an interaction between a system and an environment with a large number of degrees of freedom so that no more interference effects can be observed by practical means. Accordingly we do not invoke the notion of collapse in the sense of Copenhagen interpretation involving a discontinuous change of the wavefunction nor do we assume any gravitationally induced collapse as proposed by Penrose. The manuscript only refers to ‘collapse’ when discussing Penrose’s proposal and uses ‘effective collapse’ otherwise.

Thus, it appears to me that through their postulate the authors grant the non-local wave function itself a real character attribute. This and equating consciousness with the formation of coherences appears questionable, as the mere formation is preceding interactions with the system and the concept is basis-dependent, whereby the "preferred" basis might not be established, or multiple preferred bases (coherent and incoherent ones) might be equally relevant if several mutually exclusive "following paths" existed. For example, consider a double-slit experiment as a form of proxy of a consciousness-generating neuronal quantum process. Referring the two outcomes, particle passed through slit 1 and 2, as |1> and |2>, respectively, the particle will be in state 1/sqrt(2)(|1> + |2>) prior to passing the slits, thereby, according to the authors potentially contributing to a conscious experience. However, how could the state prior to its collapse/interaction/etc. feedback to the experiencing system and do so in a way that even permits agency through mere existence? It seems to me that the authors proposition "conscious experience arises whenever a quantum mechanical superposition forms" violates causality. What if another processes utilizes a "rotated" analyzer, thereby measuring alternative superpositions of states, thus proposing an alternative decomposition of the initial state, e.g. in the form of |1> = 1/2(|1> + |2>) + 1/2(|1> - |2>) = 1/sqrt(2)(|a> + |b>)? Dose this state, coherent in |a>,|b> but incoherent in |1>, contribute to a conscious experience, or not? With these fundamental questions unexplained, or at least incomprehensible to me, I do not see how the proposition can be considered publication ready. Please expand on the fundamental idea beyond keywords.

This is a very good point and we discussed this amongst ourselves prior to submitting. Indeed, if we say that a superposition represents distinct worlds then with respect to which basis are the superpositions defined. As the referee states, this is known as the “preferred basis” problem (Wallace, The Emergent Multiverse, 2012). But this problem has been solved! The manuscript did cite Zurek who suggested that environmental interaction selects a preferred pointer basis. But based on the comments of the referee we reviewed the literature once again and appreciated that there are several proposals to choose from. A definite algorithm for selecting an “interpretation basis” (his term for “preferred basis") has been proposed by David Deutsch in 1985 (Deutsch, Quantum Theory as a Universal Physical Theory, 1985). Deutsch in fact discusses an example similar to the one the referee constructs and concludes that this does not pose a problem for the many worlds interpretation. Other proposals for addressing the preferred basis problem are by Bell (1985) and Lockwood (1989). 

We agree with the referee that we need to make our assumption that a preferred basis has been selected more explicit. We added text and references in the section startings at line 89 and 107 to address this.

The authors suggest a range of quantum biological experiments, which sound interesting, not necessarily to support their hypothesis, but on the simpler level of exploring if quantum entanglement plays a role in consciousness. The idea appears to be to expand the mind through coupling to a quantum computer and turning the mind off using general anesthesia. While the idea is nice, the practical realizability appears to be entirely left out. Still, the suggestion is formulated as a grant proposal (which, so I assume, was the original purpose of this manuscript), as if this could be attempted tomorrow. However, details or concrete ideas are glaringly absent.  In any case, it appears that the coupling to a quantum devise can only be successful if the neuronal qubits can couple to the device. This however requires a focus on a specific realization of qubits, which the authors leave open. Nuclear spins are mentioned, but then electron spin resonance is suggested to be employed and radical pairs are mentioned, thereby putting focus on electron spins. The manuscript would profit greatly if the authors explored possible implementations of their idea in terms of the various suggestions in the literature (or alternatives).

As the manuscript states we agree that coherently coupling degrees of freedom in the brain to a quantum computer is technically very challenging. Whether such a link will achieve the desired effect of expanding conscious experience depends on our conjecture being correct and identifying biological qubits playing the part assigned to them in our conjecture. In fact the paper is structured such as to suggest simpler warm-up experiments that will teach us whether and which biological qubits may participate and how to couple external qubits to them. While initially it was dismissed outright that coherence on any relevant time and spatial scales needs to be considered when modeling the functioning of the nervous system (see e.g. Tegmark 1999), the field of quantum biology has since made progress in gathering evidence that functional quantum effects may be present (for instance in bird magnetoreception, photosynthesis or the olfactory system). In our paper we refrained from speculating what the neurobiological biological qubits may be that could implement our conjecture because the suggested experiments are intended to eventually shed light on this very question. But based on the referee's suggestion we decided to add language that describes proposals that have been made in the past: microtubules (Penrose and Hameroff), nuclear spins (Fisher), aromatic rings (Hameroff). 

It is also unclear how Xe would suppress the generation of superpositions, but at the same time enrich the conscious experience by contributing additional qubits if the nuclear spin is non-zero.

This idea was espoused in the paper by Li et al. The authors observed that xenon isotopes with half-integer spin had reduced potency relative to their nuclear spin zero counterparts. Like us Li et al.  make the assumption that quantum mechanical superpositions play a role in the formation of conscious experience. Hence they conjecture that the additional nuclear spins may aid the formation of superposition states, akin to how putting up additional antennas aids a communication network, thereby counteracting the anesthetic mechanism, whatever that mechanism may be.  

The authors assume that a quantum system comprising about 100 qubits will preclude the prediction of behavior. While their argument is true in a simplistic way, such large systems will increasingly lose their quantum traits and appear classical to the observer, permitting a treatment in terms of classical statistical physics. In fact, even the much smaller biological qubits that have been suggested so far lose coherence/entanglement on timescales that are fast relative to human cognition/perception and do so the fast the larger they are. This suggests that truly we are at most able to consider independent local systems of a few degrees of freedom. Assuming that these systems signal "0" or "1" in some form with equal probability, almost all outcomes will have a fraction 0.5 of qubits being “1”, permitting a classical model.  

The referee will probably agree with us that it ought to be uncontroversial that quantum processes involving at least a hundred qubits are present inside molecules in nerve cells, for instance inside the electronic hulls of proteins and even within nuclei. Classical states resulting from such processes are not predictable even probabilistically. If the resulting states are subjected to classical dynamics, even nonlinear dynamics, then this may often permit a classical model as the referee states and we agree with this. But not always! Think of this technical setup as an illustration: Use a bit string measured from a quantum circuit as a seed vector for a diffusion neural network trained to produce images. The resulting distribution of images is outside of our capability to model accurately. We may speculate that such a process could underlie the generation of mental imagery during dreaming. But our point is less to argue how relevant this fundamental indeterminism, also referred to as Knightian uncertainty, is behaviorally but rather to make the point in principle: the behavior of an organism will never be perfectly predictable. Until very recently this question was still regarded as open (see talks by Scott Aaronson).

We made additional small edits throughout the text.

Reviewer 2 Report

Comments and Suggestions for Authors

The paper titled “Testing the conjecture that quantum processes create conscious experience” explores the intriguing hypothesis that conscious experience arises from quantum superpositions, proposing a series of quantum biology experiments to investigate this conjecture. By linking brain tissues to quantum processors, the authors aim to uncover the role of quantum effects in shaping conscious experiences. The paper suggests that the structure of quantum superpositions determines the qualia of conscious experiences and quantum entanglement may resolve the binding problem in consciousness. The proposal also hints at the potential expansion of human conscious experience through brain-quantum computer interfaces, offering a novel approach to understanding the relationship between matter states and consciousness. The authors also suggest investigating anesthesia, particularly the effects of xenon isotopes on consciousness and anesthetic potency, due to the intriguing implications of quantum processes in altering states of consciousness.

I am impressed by the authors list of the paper including Hartmut Neven, Peter Read, Kenneth S. Kosik, Tjitse van der Molen, Dirk Bouwmeester, Eve Bodnia, Luca Turin, and Christof Koch. Among these great authors, Christof Koch is a well-known neuroscientist recognized for his work on the neural correlates of consciousness and his contributions to the field of computational neuroscience. He has published extensively on topics related to consciousness, brain function, and artificial intelligence. Koch co-authored a paper with Klaus Hepp in Nature in 2006 suggesting that quantum effects may not play a significant role in the brain, however, his current work on the topic of quantum processes and consciousness may represent a shift in focus or a new perspective based on evolving research and advancements in the field. Hartmut Neven is a notable scientist and the leader of Google Quantum AI who is known for his works on quantum machine learning and quantum supremacy, pushing the boundaries of what is possible with quantum computing. As a key author in the paper discussing the conjecture that quantum processes create conscious experience, Neven brings his expertise in quantum computing to explore the potential link between quantum phenomena and consciousness. His involvement in this interdisciplinary research project adds credibility and expertise to the discussion of quantum effects on conscious experiences. Dirk Bouwmeester is a prominent physicist known for his research in quantum optics and quantum information science. Bouwmeester collaborated with Roger Penrose and others on a paper published in Physical Review Letters (PRL) in 2003, a pioneering paper on Optomechanics, exploring the superposition of a mirror assumed to be the size of a tubulin in a microtubule. This study probably aimed to provide evidence for quantum superposition states in microtubules, potentially supporting the Orchestrated Objective Reduction (Orch OR) theory proposed by Penrose and Stuart Hameroff. By investigating quantum properties at the scale of microtubules (which are macroscopic scales), the research tries to make a connection between quantum phenomena in biological systems and consciousness, contributing to the ongoing exploration of quantum effects on cognitive processes and the physical basis of consciousness. In addition, Luca Turin's expertise in biophysics and his unique perspective on the role of quantum phenomena in biological systems could offer valuable insights. Turin's research focuses on the molecular basis of smell and the role of quantum processes in olfaction. His work on the vibrational theory of olfaction suggests that molecular vibrations play a crucial role in how we perceive scents, challenging traditional models of smell based solely on molecular shape. The collaboration of experts from diverse disciplines in this paper adds credibility and depth to the exploration of quantum processes and conscious experience. This paper certainly merits publication due to its innovative and thought-provoking approach to investigating the conjecture that quantum processes create conscious experience. Strong aspects of the paper include its innovative approach to exploring the quantum nature of consciousness, the clear articulation of the proposed experiments, and the potential implications for expanding human experiences.

However, critical questions arise regarding the ability of the experiments to establish a causal link between quantum superpositions and consciousness, the consideration of confounding variables that may influence outcomes, and the reproducibility and generalizability of the results. Addressing the following questions ensure the robustness and validity of the claims made in the paper:

-            - Can the proposed experiment effectively establish a causal link between quantum superpositions and conscious experience?

-            - How do the authors address potential confounding variables that could influence the outcomes of the proposed experiments?

-            - How do the authors ensure the reproducibility and generalizability of the results obtained from the proposed experiments?

Author Response

We want to thank the reviewer for taking the time to study our paper and for their thoughtful comments. Addressing those comments will certainly improve our manuscript.

The paper titled “Testing the conjecture that quantum processes create conscious experience” explores the intriguing hypothesis that conscious experience arises from quantum superpositions, proposing a series of quantum biology experiments to investigate this conjecture. By linking brain tissues to quantum processors, the authors aim to uncover the role of quantum effects in shaping conscious experiences. The paper suggests that the structure of quantum superpositions determines the qualia of conscious experiences and quantum entanglement may resolve the binding problem in consciousness. The proposal also hints at the potential expansion of human conscious experience through brain-quantum computer interfaces, offering a novel approach to understanding the relationship between matter states and consciousness. The authors also suggest investigating anesthesia, particularly the effects of xenon isotopes on consciousness and anesthetic potency, due to the intriguing implications of quantum processes in altering states of consciousness.

I am impressed by the authors list of the paper including Hartmut Neven, Peter Read, Kenneth S. Kosik, Tjitse van der Molen, Dirk Bouwmeester, Eve Bodnia, Luca Turin, and Christof Koch. Among these great authors, Christof Koch is a well-known neuroscientist recognized for his work on the neural correlates of consciousness and his contributions to the field of computational neuroscience. He has published extensively on topics related to consciousness, brain function, and artificial intelligence. Koch co-authored a paper with Klaus Hepp in Nature in 2006 suggesting that quantum effects may not play a significant role in the brain, however, his current work on the topic of quantum processes and consciousness may represent a shift in focus or a new perspective based on evolving research and advancements in the field. Hartmut Neven is a notable scientist and the leader of Google Quantum AI who is known for his works on quantum machine learning and quantum supremacy, pushing the boundaries of what is possible with quantum computing. As a key author in the paper discussing the conjecture that quantum processes create conscious experience, Neven brings his expertise in quantum computing to explore the potential link between quantum phenomena and consciousness. His involvement in this interdisciplinary research project adds credibility and expertise to the discussion of quantum effects on conscious experiences. Dirk Bouwmeester is a prominent physicist known for his research in quantum optics and quantum information science. Bouwmeester collaborated with Roger Penrose and others on a paper published in Physical Review Letters (PRL) in 2003, a pioneering paper on Optomechanics, exploring the superposition of a mirror assumed to be the size of a tubulin in a microtubule. This study probably aimed to provide evidence for quantum superposition states in microtubules, potentially supporting the Orchestrated Objective Reduction (Orch OR) theory proposed by Penrose and Stuart Hameroff. By investigating quantum properties at the scale of microtubules (which are macroscopic scales), the research tries to make a connection between quantum phenomena in biological systems and consciousness, contributing to the ongoing exploration of quantum effects on cognitive processes and the physical basis of consciousness. In addition, Luca Turin's expertise in biophysics and his unique perspective on the role of quantum phenomena in biological systems could offer valuable insights. Turin's research focuses on the molecular basis of smell and the role of quantum processes in olfaction. His work on the vibrational theory of olfaction suggests that molecular vibrations play a crucial role in how we perceive scents, challenging traditional models of smell based solely on molecular shape. The collaboration of experts from diverse disciplines in this paper adds credibility and depth to the exploration of quantum processes and conscious experience. This paper certainly merits publication due to its innovative and thought-provoking approach to investigating the conjecture that quantum processes create conscious experience. Strong aspects of the paper include its innovative approach to exploring the quantum nature of consciousness, the clear articulation of the proposed experiments, and the potential implications for expanding human experiences.

 However, critical questions arise regarding the ability of the experiments to establish a causal link between quantum superpositions and consciousness, the consideration of confounding variables that may influence outcomes, and the reproducibility and generalizability of the results. Addressing the following questions ensure the robustness and validity of the claims made in the paper:

- Can the proposed experiment effectively establish a causal link between quantum superpositions and conscious experience?

A robust, reproducible effect of isotopic Xenon on anesthesia (threshold in the case of Drosophila, neural activity in the case of organoids) would likely be impossible to account for by any classical effect. Demonstrating such an isotope effect would in essence kick-start the search for the quantum mechanism responsible. We will be exploring the theoretical basis behind xenon influence on organoid activity using molecular simulation tools developed at Allen Brain Institute with help Anton Archipov and Christof Koch.

- How do the authors address potential confounding variables that could influence the outcomes of the proposed experiments?

Confounding variables will likely be specific to each experiment and preparation and cannot be addressed in a general fashion. We will address confounding variables via standard tools of statistical analysis. The statistical measure will come from analyzing differences between the organoid within the same batch and differences of observed signal in organoid activities as a function of xenon isotopes, doses and pressure. From the data analysis side, we compare the results from the real data to the constrained randomness- shuffled model based on results of Michael Okun et al (The Journal of Neuroscience, 2012), which allows us to extract its statistical significance. From the experimental side, we compare statistics of the organoid cell compounds based on the results of single RNA sequencing and  immunocytochemistry (ICC). We will proceed in an analogous manner for the fruit fly experiments.

 - How do the authors ensure the reproducibility and generalizability of the results obtained from the proposed experiments?

General anesthesia is the most general phenomenon in all of neuroscience. All animals from Paramecium to man are rendered immobile and unresponsive by the same set of general anesthetics. Our experiments involve two preparations at the extremes of this spectrum: brain organoids and live fruit flies. We are confident, given our knowledge of the spectrum of action of general anesthetics, that Xenon effects on organoids and fruit flies will generalize (naturally with some changes in dosage and possible relative potencies) to other animal species. We are primarily interested in differences in response between one Xe isotope and another. The design of experiments will be adapted to each of the preparations to ensure adequate statistics are obtained. To collect statistics from organoids, we need to ensure the cell culture composition is the same for the whole organoids batch. We will be using brain organoids based on the Pasca protocol, which are better reproducible in contrast to Lancaster protocol (see Steven A. Sloan et all (Nat Protoc. 2018), works of Sergiu P. PaÈ™ca  and all people who cite this work). The brain organoids composition will be obtained via single RNA cell sequencing or/and immunocytochemistry (ICC) analysis.

We made additional small edits throughout the text.

Round 2

Reviewer 1 Report

Comments and Suggestions for Authors

The authors have made sufficient changes to their manuscript, allowing me to follow their idea. While, evidently, the proposal is highly speculative and I am sceptical, I see no error in publishing the manuscript as a discussion piece.

Author Response

Thanks

Reviewer 2 Report

Comments and Suggestions for Authors

My recommendation is accepting the paper as they have addressed all the comments.

Author Response

Thanks